# Serological Evidence of Lassa Virus Exposure in Non-*Mastomys* Small Mammals Within a Hyperendemic Region of North-Central Nigeria: A Pilot Study

**DOI:** 10.3390/v17101368

**Published:** 2025-10-13

**Authors:** Augustine Ovie Edegbene, Temidayo Oluwatosin Omotehinwa, Joseph Anejo-Okopi, Sara El Yaagoubi, Oladapo Sunday Shittu, Onyemocho Audu, Evangeline Olohi Abah, Samuel Ijoganu, Genesis Kwaghgande, Celina Aju-Ameh, Adesanya Abimbola, Emmanuel Otache, Emmanuel Ameh, Joyce Danyi, Owoicho Ikwu, Esther Agmdalo Malachi Cegbeyi, Oludare Oladipo Agboola, Joseph Okoeguale, Reuben Agbons Eifediyi, Ediga Bede Agbo, John Alechenu Idoko, Innocent Otoboh Achanya Ujah, Stephen Obekpa Abah

**Affiliations:** 1Department of Biological Sciences, Federal University of Health Sciences, Otukpo 972261, Nigeria; 2Institute for Global Health and Health Security, Federal University of Health Sciences, Otukpo 972261, Nigeria; 3Department of Mathematics and Computer Science, Federal University of Health Sciences, Otukpo 972261, Nigeria; 4Department of Microbiology, Federal University of Health Sciences, Otukpo 972261, Nigeria; 5LESCB URL-CNRST N° 18, Faculty of Sciences, Abdelmalek Essaâdi University, Tétouan, Morocco; 6Department of Obstetrics and Gynecology, College of Medicine, Federal University of Health Sciences, Otukpo 972261, Nigeria; 7Department of Community Medicine, College of Medicine, Federal University of Health Sciences, Otukpo 972261, Nigeria; 8Department of Physiology, College of Medicine, Federal University of Health Sciences, Otukpo 972261, Nigeria; 9Department of Nursing Services, Benue State University Teaching Hospital, Makurdi 970101, Nigeria; 10Federal Teaching Hospital, Lokoja 260101, Nigeria; 11Department of Community Medicine, College of Medicine, University of Jos Teaching Hospital, Jos 930103, Nigeria; 12Federal Medical Centre, Keffi 961101, Nigeria; 13Palmcrest Ent Specialist Hospital, Abuja 900001, Nigeria; 14University of Abuja Teaching Hospital, Abuja 900001, Nigeria; 15Department of Botany, Federal University, Lokoja 260101, Nigeria; 16Institute of Viral and Emergent Pathogens Control and Research, Irrua Specialist Teaching Hospital Irrua, Irrua 310120, Nigeria; 17Faculty of Clinical Sciences, Ambrose Alli University Ekpoma, Ekpoma 310101, Nigeria; 18Department of Pediatrics, University of Jos, Jos 930103, Nigeria

**Keywords:** non-*Mastomys* small mammals, *Mastomys natalensis*, Lassa virus (LASV), Lassa fever, rodent reservoirs, One Health, Nigeria, West Africa

## Abstract

Lassa fever (LF), a severe hemorrhagic disease endemic to West Africa, is primarily transmitted by rodents of the genus *Mastomys*, particularly *Mastomys natalensis*, which serve as the main reservoirs of Lassa virus (LASV). There have been reports of high prevalence of LF in Nigeria, and outbreaks tend to be recurrent yet geographically restricted, implying that additional ecological or epidemiological factors influence the distribution of the disease beyond the mere presence of *M. natalensis*. However, national-scale data on LASV prevalence in rodent populations remain scarce. To address this gap, a targeted small mammal survey was conducted over a four-month period (May to August 2024) in Otukpo Local Government Area (LGA) of Benue State, north-central Nigeria. Rodents and other small mammals were trapped across three purposively selected wards identified as high-risk areas based on prior reports of occurrence of such small mammals in the areas and the informal settlements in which the selected wards were located in in Otukpo LGA. Analysis of the samples revealed no statistically significant variation in LASV prevalence among the study sites, indicating a relatively uniform, low-level exposure risk across the LGA and region. However, a marginally significant difference in LASV detection between plasma and serum samples suggests that sample type and storage conditions may influence serological sensitivity. These findings highlight the importance of refining diagnostic protocols, broadening surveillance to include additional rodent hosts, and integrating ecological data with public health strategies to improve early warning systems and strengthen Lassa fever control efforts.

## 1. Introduction

Lassa fever (LF) is a severe acute viral hemorrhagic illness caused by the Lassa virus (LASV), an arenavirus endemic to several West African countries, including Nigeria, Sierra Leone, Guinea, and Liberia [1,2,3]. Clinically, the disease presents a wide spectrum of manifestations, ranging from mild, non-specific symptoms to severe hemorrhagic manifestations often associated with multiorgan failure [4,5,6,7]. LASV, the causative agent of LF, is a rodent-borne arenavirus and a significant zoonotic pathogen responsible for an estimated 5000 deaths annually in West Africa [5,8]. Human infection typically results from direct or indirect contact with the urine or feces of infected rodents, primarily the multimammate rat *Mastomys natalensis*, which is historically recognized as the principal reservoir of LASV [9,10]. These rodents are widely distributed across sub-Saharan Africa and have also been reported in North Africa, including Morocco [11]. However, recent phylogenetic and ecological studies have challenged this single-host model, revealing that multiple *Mastomys* species maintain LASV—and potentially other rodent and small mammal hosts—thereby highlighting the complex and multifactorial nature of the virus’s transmission ecology [3,10,12,13,14]. This complexity is further compounded by the unresolved taxonomy of *Mastomys*, a genus that comprises at least eight distinct species, with differentiation based in part on diploid chromosome numbers of 32, 36, and 38, and this also further contributes to the challenges in accurately identifying reservoirs and understanding host-specific dynamics [15,16].

Ecologically, *Mastomys* species are highly adaptable, occupying a wide range of habitats including grasslands, savannahs, croplands, forest edges, and rural dwellings [17,18,19]. These rodents are nocturnal and among the most fecund of small mammals, with high reproductive rates that contribute to their ecological success. While the dynamics of natural LASV infection in *Mastomys* populations remain incompletely understood, laboratory studies have provided important insights into their infection biology. For instance, juvenile *Mastomys* with a diploid number of 36 (2*n* = 36), when experimentally infected intraperitoneally within the first weeks of life, can become chronic carriers, maintaining persistent viremia and viruria [20,21]. In contrast, adult individuals typically develop only transient viremia, followed by robust protective immunity. Notably, despite persistent infection, LASV does not appear to cause overt disease in *Mastomys*, although histopathological changes have occasionally been documented [22]. Reproductive output in infected females also appears unaffected, as litter sizes are comparable to those of uninfected controls [21]. Hence, the epidemiological dynamics of LF are quite complex due to these multifaceted routes of infection among varied species of small mammals as well as sex-based (male or female) response of the virus.

Epidemiological evidence of human Lassa fever is most extensively documented in Nigeria, Liberia, Sierra Leone, and Guinea, as demonstrated by numerous clinical and field investigations [2,7,17,23,24,25]. These countries represent the core endemic zones where the disease burden is the highest, with an estimated 300,000 to 500,000 cases and approximately 5000 deaths annually [7,26,27]. As mentioned earlier, the causative agent of the virus belongs to the family Arenaviridae, genus Arenavirus, and is naturally harbored by the multimammate rat (*Mastomys natalensis*), recognized as the primary reservoir host [7,9,28,29,30]. Although *M. natalensis* is widely distributed across sub-Saharan Africa, LASV-infected populations have been predominantly identified in specific West African regions, particularly Nigeria, Sierra Leone, and Guinea [9,28,31,32], confirming the endemism of this virus in these countries, and Nigeria, in particular, is the most populated country in the region and Africa as a whole, so coordinated efforts are needed to curtail the continued spread of the virus.

Beyond these well-established endemic areas, antibody prevalence surveys and anecdotal reports have revealed evidence of LASV or related arenavirus circulation in other West African countries, indicating to a potentially wider distribution than currently confirmed [33,34,35,36,37]. Serological findings have also emerged from parts of Central Africa, reinforcing the need for broader surveillance and enhanced diagnostic capacity across the region [9,38]. Furthermore, sporadic imported cases of LF have been reported in Europe and North America, primarily involving travelers returning from endemic areas. These incidents underscore LASV’s significance as a globally relevant viral hemorrhagic fever and highlight the importance of international preparedness and response capabilities [38,39,40], which must be taken very seriously as further spread of the virus resulting from these imported cases can translate into a global pandemic if care is not taken to curtail the spread.

In Nigeria, particularly in the north-central region where LF outbreaks are frequently reported, the role of non-*Mastomys* rodents in the maintenance and transmission of LASV remains largely understudied [41,42]. While *Mastomys natalensis* has traditionally been identified as the primary reservoir, growing evidence suggests that other small mammal species may also contribute to the persistent spread of LASV. A comprehensive understanding of the full spectrum of potential host species is crucial for accurately predicting the dynamics of disease emergence, enhancing epidemiological surveillance, and informing the development of more effective and targeted public health interventions [43,44]. Hence, the current study in North-central Nigeria helps us unravel the multiple host dynamics of the virus, in preparation for tackling the disease and preventing national emergencies resulting from its spread.

Recent studies have provided valuable insights into the occurrence of arenaviruses among small mammals in Nigeria. Specifically, LASV lineage II and a novel Mobala-like virus have been identified in the Natal multimammate mouse (*Mastomys natalensis*). In contrast, LASV lineage III has been detected in the Guinea multimammate mouse (*Mastomys erythroleucus*) and the Kako strain of LASV in the African wood mouse (*Hylomyscus pamfi*) [10,45,46]. The use of antigen and antibody screening, particularly for immunoglobulin G (IgG), has proven effective in detecting both acute and past infections, thereby offering a clearer picture of arenavirus occurrence and prevalence patterns. Consequently, the present study aimed to integrate PCR-based detection results, as have earlier been employed by Olayemi et al. [10,45] and Redding et al. [47], with newly acquired serological data, which are generated from the same individual animals, to enhance our understanding of arenavirus circulation in Nigerian small mammal populations.

Furthermore, recent research in Nigeria has challenged the long-standing view that *Mastomys natalensis* is the sole reservoir of Lassa virus (LASV), revealing a more complex transmission ecology involving multiple rodent hosts [42]. LASV RNA and antibodies have been detected in several non-*Mastomys* genera, including *Rattus rattus*, *Hylomyscus pamfi*, and *Tatera* spp., with *R. rattus* showing positivity rates as high as 77.3% in some studies [42]. These findings suggest that such species may function either as independent reservoirs or as secondary hosts due to habitat overlap and ecological interactions with *M. natalensis*. Their frequent detection in domestic and peridomestic settings underscores the need to investigate their roles in LASV maintenance and transmission. Understanding the involvement of these alternative hosts is critical for refining rodent control strategies beyond *M. natalensis*, identifying new transmission pathways in high-risk human–rodent contact zones, and improving outbreak prediction models amid ongoing climate and land-use changes. However, key knowledge gaps remain, particularly regarding whether these species sustain LASV through vertical or horizontal transmission or act merely as spillover hosts. To address these issues, this study aims to: (i) identify and classify small mammal species (rodents and rats) inhabiting Lassa fever virus in selected endemic communities in North-Central Nigeria; (ii) detect and characterize the presence of LASV RNA and antibodies in non-*Mastomys* rodent populations using molecular (PCR) and serological (IgG) approaches; and (iii) assess the potential role of these non-*Mastomys* rodents in LASV transmission through integrated ecological and virological analyses. These objectives are designed to fill critical gaps in our understanding of LASV ecology and to test the hypothesis that non-*Mastomys* species may significantly contribute to LASV maintenance and transmission cycles in endemic regions.

## 2. Materials and Methods

### 2.1. Ethical Approval

Ethical approval was obtained from the Ethical Committee of Federal University of Health Sciences, Otukpo (FUHSO), Nigeria, and the National Health Research Ethics Committee (NHREC) of Nigeria with approval number: NHREC 1 January 2007–1 September 2023.

### 2.2. Study Sites and Sampling Technique

This study was conducted from May to August 2024 in Otukpo Local Government Area (LGA), Benue State, North-Central Nigeria (see Figure 1). The LGA has an estimated population of 370,000 [Source: National Population Commission, 2006 projection] and lies within the Guinea savanna ecological zone, characterized by a tropical climate with distinct wet and dry seasons, average temperatures of 25–29 °C, and 68% humidity [48].

A multistage sampling technique was employed. In the first stage, three wards (Ewulo, Otukpo Central, and Otukpo Town East) were purposively selected based on prior reports from local health authorities indicating these as high-risk areas for Lassa fever outbreaks and rodent activity. These wards are predominantly peri urban environments. In the second stage, three communities were randomly selected from each ward: Ewulo Ward; Igbanonmaje (latitude: 7.204993, longitude: 8.148024), Otada (latitude: 7.226118, longitude: 8.14701), and Upu (latitude: 7.215542, longitude: 8.150755). In Otukpo central ward: Timber Depot 1 (latitude: 7.199372, longitude: 8.140525), Timber Depot 2 (latitude: 7.199690, longitude: 8.139603) and Otukpo main market axis (latitude: 8.131695, longitude: 7.201515). In Otukpo town east, the following were selected: Tiv market axis (latitude: 8.151438, longitude: 7.199933), Ogiri Oko street (latitude: 8.138123, longitude: 7.194729) and General Hospital axis (latitude: 7.202721, longitude: 8.150125).

Within each community, sampling locations (e.g., individual households, dumpsites, market stalls) were identified based on community compliance and signs of rodent activity. In total, 150 such locations were mapped, and fabricated life traps were deployed at each.

The sample size was calculated using the standard formula for estimating a single population proportion in a cross-sectional study [49]. A conservative expected seroprevalence (p) of 5% was assumed. This conservative estimate was chosen based on the lower bound of reported positivity rates in non-*Mastomys* rodents from a recent study in Nigeria [42], which found much higher rates (e.g., up to 77.3% in *Rattus rattus*), to ensure a sufficient sample size even if the true prevalence was low. For a 95% confidence level (Z = 1.96) and a desired precision (d) of 5%, the initial sample size (n) was calculated as follows:

n = [Z^2^ × p × (1 − p)/d^2^]


This calculation yielded a minimum of 73 animals. After adjusting for a 20% non-response rate (e.g., trap avoidance, empty traps), the target sample size was increased to 92.

### 2.3. Traps Fabrication

Live-capture box traps were fabricated locally to ensure cost-effectiveness and adaptability to field conditions. The traps were designed with three slightly different internal dimensions (L × W × H) to accommodate various small mammal sizes: (i) 30 × 16.5 × 12 cm, (ii) 37 × 15 × 13.5 cm, and (iii) 30 × 18.2 × 14 cm. An example of the fabricated live capture trap used in the study is presented in Figure 2. This multi-size approach aimed to optimize capture efficiency across different species. All traps were constructed from durable, locally sourced materials, primarily condemned fridge steel and wood, ensuring they were humane, reusable, and provided sufficient ventilation.

The fabrication process was closely supervised by a minimum of three research team members under the supervision of Dr. Augustine Ovie Edegbene at all times to ensure consistency, structural integrity, and that the design specifications—particularly a secure but non-injurious closing mechanism—were strictly adhered to. The use of live-capture traps was a deliberate choice to allow for the collection of biological samples (e.g., blood) from live animals, which is essential for accurate virological and serological testing.

### 2.4. Trapping of Small Mammals

In each community, two local field workers were recruited to deploy the fabricated traps at designated locations. Traps were placed in strategic locations known to be rodents’ pathways including houses, bush paths, at dumpsites and near pit latrines. Trap density varied by location based on available space and homeowner compliance. The geographic coordinates (latitudes and longitudes) of all deployment locations within each community were mapped out for easy tracing and proper allocation of captured small mammals to a given location. All traps set were rebaited every morning or immediately the trap catches rodent(s) or rat(s). Deployed traps were retained for a period of up to 10 days per location and were checked by the local field workers twice daily—morning and evening—to retrieve the captured small mammals. Baits used included crayfish, smoked fish, cassava, roasted yams, maize and groundnut. The total number of traps set per location (e.g., houses, pit latrines, dumpsites) varied between 80 and 210, depending on the number of households willing to trap placement. Capture success of small mammals was calculated following the protocol described by Demby et al. [19] as cited in Happi et al. [42]. Trap success for a location was defined as the total number of captures in a location divided by the number of traps set each night and the number of trap nights, multiplied by 100.

### 2.5. Identification, Dissection, and Sample Collection of Small Mammals

All small mammals captured in the field were transported to the laboratory for morphological measurement and biological sample collection. Captured animals were first sedated with chloroform by placement in a desiccator for three minutes prior to subsequent procedures, including species identification, morphological measurement, and blood collection.

Morphological parameters measured included body weight, head-body length, tail length, hind foot length, and ear length. Other recorded parameters included sex, eye color, fur color, and mammary gland arrangement and count (in mature females). Primary species identification was performed by a trained zoologist on the research team using standard taxonomic keys and morphological measurements, aided by pictorial guides of Nigerian small mammals. The iNaturalist application was used for preliminary identification support.

Following morphological measurement, blood samples were collected via cardiac puncture from sedated animals. The thoracic cavity was accessed using dissecting scissors, and blood was drawn from the heart using a 2 mL syringe. The blood samples were dispensed into EDTA and plain tubes to obtain plasma and sera, respectively.

Plasma and serum were separated by centrifugation at 1600 RPM at the Infectious Disease Laboratory of the Federal University of Health Sciences Teaching Hospital, Otukpo (FUHSOTH). The separated components were transferred to cryovial tubes and temporarily stored at −80 °C before shipment to the Institute of Viral Hemorrhagic Fever and Other Emergent Pathogens (IVEP), Irrua Specialist Teaching Hospital, Edo State, for serological and PCR testing.

### 2.6. Serological Testing of the Blood Samples for Lassa Fever Virus (LFV) Antibodies

Serum samples isolated from the captured small mammals were screened for Lassa virus (LASV) antibodies using an indirect enzyme-linked immunosorbent assay (ELISA), following standard protocols adapted from established methods for arenavirus surveillance in rodent reservoirs [10,50,51]. The ELISA was performed using recombinant LASV nucleoprotein (NP) antigen from lineage II, which is the predominant lineage in Nigeria, to ensure optimal detection sensitivity for antibodies against locally circulating viruses.

Briefly, 96-well microplates were coated with 100 µL per well of purified LASV antigen and incubated overnight at 4 °C under Biosafety Level-3 (BSL-3) conditions. The plates were then washed with phosphate-buffered saline containing 0.05% Tween-20 (PBS-T) and blocked with 200 µL of blocking buffer (5% bovine serum albumin) at 37 °C for 1 h to prevent non-specific binding.

Rodent serum samples were diluted 1:100 in sample diluent, and 100 µL of each sample was added in duplicate to the wells alongside known positive and negative controls. After incubation at 37 °C for 1 h, plates were washed thoroughly, and 100 µL of horseradish peroxidase (HRP)-conjugated anti-rodent IgG or IgM antibody was added to detect bound immunoglobulins. Following an additional 1-h incubation at 37 °C, plates were washed and developed with 100 µL of tetramethylbenzidine (TMB) substrate for 20 min in the dark. The reaction was stopped with 1N sulfuric acid, and the optical density (OD) was measured at 450 nm using a microplate reader.

A sample was considered positive if its mean OD value exceeded the mean OD of the negative controls plus three standard deviations, in accordance with standard cut-off criteria [50]. All assays were performed in duplicate and validated using internal quality controls included on each plate.

### 2.7. Extraction of RNA and Reverse Transcription PCR (RT-PCR)

Total RNA was extracted from rodent blood samples to detect active Lassa virus (LASV) infection, following protocols adapted from Fichet-Calvet et al. [50], Bowen et al. [52] and Cheng et al. [53]. All extractions and subsequent manipulations were performed under Biosafety Level-3 (BSL-3) conditions with appropriate personal protective equipment to minimize the risk of laboratory-acquired infection.

For the extraction, 100 µL of whole blood was homogenized in 600 µL of lysis buffer provided in the QIAamp Viral RNA Mini Kit (Qiagen, Hilden, Germany) according to the manufacturer’s instructions. Homogenization was performed using a sterile mechanical tissue grinder. Lysates were incubated with proteinase K at 56 °C for 10 min and then processed according to the spin-column protocol, including on-column DNase I treatment to remove potential genomic DNA contamination. Viral RNA was eluted in 60 µL of RNase-free water and stored at −80 °C until use.

Detection of LASV RNA was performed by one-step reverse transcription PCR (RT-PCR) targeting the nucleoprotein (NP) gene, which is conserved across LASV lineages [52]. Primers targeting the highly conserved NP gene (LVNP-F: 5′-GGAAGAGGCAACTGGCATAT-3′ and LVNP-R: 5′-CTCCATGTTGTTGAGGTTGT-3′) were used, as described by Olayemi et al. [10] and Bowen et al. [52]. This assay has been validated to detect a broad range of LASV lineages endemic to Nigeria.

Reactions were carried out using the SuperScript™ III One-Step RT-PCR System with Platinum^®^ Taq DNA Polymerase (Invitrogen, Carlsbad, CA, USA). Each 25 µL reaction mixture contained 12.5 µL of 2× reaction buffer, 0.5 µL of SuperScript III RT/Platinum Taq Mix, 400 nM each of forward and reverse primers specific for the LASV NP gene, 1 µL of extracted RNA template, and nuclease-free water to volume.

Thermal cycling was performed in a GeneAmp PCR System 9700 (Applied Biosystems, San Francisco, CA, USA) under the following conditions: reverse transcription at 50 °C for 30 min; initial denaturation at 94 °C for 2 min; 40 cycles of denaturation at 94 °C for 30 s, annealing at 55 °C for 30 s, and extension at 68 °C for 1 min; with a final extension at 68 °C for 5 min.

PCR products were resolved by electrophoresis on a 1.5% agarose gel stained with ethidium bromide and visualized under UV illumination [54]. Amplicons of the expected size (~300 bp) were excised and purified for Sanger sequencing to confirm LASV-specific sequences. Positive and negative extraction controls were included in each batch to monitor for cross-contamination and false-positive results.

### 2.8. Data and Statistical Analyses

The results are presented in tables and charts (bar and pie charts). Datasets were subjected to various statistical tests such as one-way analysis of variance (ANOVA), Monte-Carlo permutation, t-test, and chi-square test. *p* values greater than 0.05 were considered statistically not significant. Student’s t-test was used to project the level of significance in the survival rate between the males and females’ small mammals captured alive. One-way ANOVA was used to ascertain the statistical significance difference in the capture rate vs. survival rate of small mammals among the nine communities sampled in the three wards sampled in Otukpo. A one-way ANOVA was also used to show the level of statistical significance in the number of positive cases of LFV antibodies in the blood samples tested among the nine communities sampled in Otukpo LGA. We conducted a Chi square test to ascertain the level of significant difference in the prevalence rate of LF infection between the two blood sample types for both IgG and IgM, while a Monte Carlo permutation test was used to confirm the level of significance in the prevalence rate of LF infection in the blood sample types. All charts and statistical tests were constructed and conducted using the paleontological statistical package (PAST; version 16.0).

## 3. Results

### 3.1. Live Capture Rate vs. Survival Rate of Rodents and Rats in Otukpo LGA, North-Central Nigeria

Figure 3i–iii present the number of rodents captured alive and subsequently delivered viable to the laboratory. In Ewulo ward, 40 live captures were recorded (Figure 3i), distributed across the communities as follows: Igbanonmaje (33), Otada (4), and Upu (3). Of these, 26, 4, and 3 animals were delivered alive to the laboratory from Igbanonmaje, Otada, and Upu, respectively.

In Otukpo Town Central ward, 92 live captures were recorded (Figure 3ii), distributed as follows: Timber Depot 1 (58), Timber Depot 2 (14), and Otukpo Main Market axis (20). Subsequently, 24, 11, and 16 live animals were successfully delivered from these locations, respectively.

In Otukpo Town East ward, a total of 25 live captures of rodents and rats were recorded, 16 in Tiv market axis, 4 in Ogiri Oko street, and 5 in General Hospital axis (Figure 3iii). Of the 16 live captures of rodents and rats at Tiv market, only 5 were delivered to the laboratory alive. Of the 4 live captures at Ogiri Oko street, only 1 was delivered to the laboratory alive, while 2 delivered alive from the 5 live captures in General Hospital axis (Figure 3iii).

Overall, 157 rodents were captured alive across all sites, of which 92 (58.6%) were successfully delivered alive to the laboratory. A one-way analysis of variance (ANOVA) indicated a statistically significant difference in survival rates among the nine communities (F = 0.0047, *p* < 0.05). Of the 92 animals delivered alive, 65 (70.7%) were male, and 27 (29.3%) were female. A Student’s t-test found no significant difference in survival rates between sexes (t = 0.067, *p* > 0.05).

### 3.2. Overview of the Capture Success of Rodents and Rats in the Nine Communities Sampled in Otukpo LGA, Benue State, North-Central Nigeria

Table 1 provides a summary of the number of individuals and capture success rates of rodents and rats recorded at each sampling location. In the Igbanonmaje community, four taxa were captured: Mus musculus with nine individuals (47.36% capture success), *Rattus rattus* with 12 individuals (63.16%), Tatera species with four individuals (10.50%), and *Pallasiomys* species with one individual (5.26%). In the Otada community, two taxa were identified: *Mus musculus* with three individuals (12.50%) and *Rattus rattus* with one individual (4.17%). At Upu community, two taxa were also recorded, comprising Mus musculus with one individual (5.00%) and *Rattus rattus* with two individuals (10.00%). For Timber Depot 1, two taxa were captured: Mus musculus with nine individuals, showing a reported capture success of 45.0%, and *Rattus rattus* with fifteen individuals (27.0%). Similarly, in Timber Depot 2, Mus musculus accounted for seven individuals (14.0%) and *Rattus rattus* for four individuals (8.0%). In the Otukpo main market axis, two taxa were found: *Rattus rattus* with fifteen individuals (37.5%) and Rattus norvegicus with one individual (18.8%). At the Tiv market axis community, only *Rattus rattus* was recorded, with five individuals and a capture success of 66.67%. In Ogori Oko Street, a single taxon, *Rattus rattus*, was captured with one individual (5.6%), while at the General Hospital Axis, two individuals of *Rattus rattus* were recorded, representing an 18.5% capture success.

### 3.3. Seroprevalence of Lassa Virus Antibodies

To ascertain the level of exposure of the captured small mammals to Lassa fever previously, we carried out serological tests on all the blood samples (87 plasma and 64 sera) extracted from the 92 live small mammals (rodents and rats). Four small mammals were reported as positives for IgG and IgM (two positives each for both IgG and IgM; Figure 4). Of the four positive samples, three were *Rattus rattus* (one IgG and two IgM), and one was Mus musculus (IgG); Figure 5.

All the four small mammals’ samples that tested positive for both IgG and IgM were males. Based on the results of serology per ward/community, Ewulo ward reported the highest number of positive samples (two from Igbanonmaje community and one from Upu community; Figure 6). The remaining one positive sample was recorded in Otukpo main market axis in Otukpo Town Central ward (Figure 5). A one-way ANOVA was performed to show that there is no significant difference in the number of positive cases of LFV antibodies in blood samples tested among the nine communities sampled in Otukpo LGA (F = 2, df = 16, *p* = 0.1765, *p* > 0.05).

Four out of 92 animals (4.3%) tested positive for LASV antibodies (IgG or IgM). Among the positive samples, three were *Rattus rattus*, and one was Mus musculus, representing a species-specific seroprevalence of 75% and 25%, respectively.

The overall antibody prevalence, based on 151 blood samples (including both plasma and serum aliquots from the 92 animals), was 2.65% (4/151). Seropositivity was higher in serum samples (4.69%, 3/64) than in plasma samples (1.15%, 1/87). A Chi-square test indicated that this difference was not statistically significant (χ^2^ = 3.29, *p* = 0.0696). This finding was corroborated by a Monte Carlo permutation test (*p* = 0.0656).

## 4. Discussion

Our study provides preliminary insights into LASV exposure among non-*Mastomys* rodents in Otukpo LGA, North-Central Nigeria, and highlights methodological challenges inherent in such field investigations.

### 4.1. Logistical Challenges and Sampling Limitations

The observed discrepancies between the live capture rates (*n* = 157) and the number of small mammals delivered alive to the laboratory (*n* = 92) across the three wards in the study area underscore significant logistical and ecological challenges in field sampling. The overall survival rate of 58.6% (92/157) reflects not only the fragility of small mammals during trapping and transport but possibly species-specific stress responses and handling-related mortality, which have been widely documented in rodent ecology and epidemiological surveillance studies [55,56]. The high mortality rate (41.4%) between capture and processing may have introduced a survival bias. Stressed or sick animals, which could have different viral loads or serological profiles, may have been less likely to survive transport, potentially leading to an underestimation of true prevalence. These survival constraints are crucial because only live specimens allow for full biological sampling (e.g., blood, tissue) and accurate pathogen detection [57]. Furthermore, the significant difference in capture vs. survival rate across the nine sampled communities (ANOVA: F = 0.0047; *p* < 0.05) suggests that environmental or anthropogenic factors (e.g., trap placement, handling methods, ambient temperature, or time between capture and processing) might vary considerably by location. Studies have shown that high ambient temperatures and prolonged confinement in traps can lead to increased rodent mortality [16], and optimizing these conditions is vital to enhance data quality in zoonotic disease monitoring. Although a sex-based analysis of survival did not reveal significant differences (t = 0.067; *p* > 0.05), the higher number of males (65) than females (27) among the survivors could reflect behavioral traits, such as greater mobility or trap susceptibility of males, which has been observed in other rodent surveillance programs elsewhere [9,58]. The significantly skewed sex ratio (70.7% male) suggests a trapping bias towards more mobile or trap-prone males. As all seropositive specimens were male, this bias may have influenced our prevalence estimate, and the results may not be fully representative of the entire rodent population. These patterns could influence population structure assessments and may bias pathogen prevalence estimates if one sex is underrepresented.

Crucially, despite targeting areas with prior reports of small mammals, *Mastomys natalensis*—the primary LASV reservoir—was notably absent from our captures, representing a major limitation. Habitat assessments indicate that many sampling locations were characterized by fragmented vegetation and high human activity, conditions that may be suboptimal for this species, which typically prefers dense shrub or grass cover near human dwellings. Additionally, the trapping protocol, which employed shearer traps primarily along transects in open areas, may have introduced a bias against capturing *M. natalensis*, given its specific microhabitat preferences. Consequently, while the absence of *M. natalensis* limits direct inference about its role in LASV circulation in these sites, the presence of other rodent species provides valuable insights into alternative hosts and the broader ecological dynamics of LASV in the region.

### 4.2. Small Mammal Community Composition

The rodent community captured in this study was dominated by the synanthropic species *Rattus rattus* and *Mus musculus*, which were present in nearly all sampling sites. Their high abundance in peri-domestic environments such as marketplaces and timber depots is consistent with findings from other West African urban settings, where food availability and shelter support the proliferation of these commensal species [59,60].

Other genera, including *Tatera* and *Pallasiomys*, were captured in low numbers, suggesting either low natural abundance in these habitats or lower capture probability due to behavioral traits such as trap avoidance or distinct microhabitat preferences [46].

Notably, *Mastomys natalensis*, the primary reservoir of LASV, was absent from all captures. This finding is ecologically significant and likely reflects the peri-urban nature of the study sites, which were characterized by intense human activity and infrastructure. This observation aligns with established literature indicating that *M. natalensis* tends to prefer rural, agricultural, and fallow land habitats and is often outcompeted by more synanthropic rodents like *R. rattus* and *M. musculus* in densely built environments [2,18]. Consequently, while this study provides insights into LASV exposure within urban rodent communities, the absence of *Mastomys* limits direct inferences about reservoir dynamics in the broader region and underscores the importance of habitat context in shaping rodent community structure and associated zoonotic risk.

### 4.3. Evidence of LASV Exposure and Implications for Reservoir Status

Serological testing revealed LASV-specific antibodies in 4.3% (4/92) of captured rodents, indicating prior virus exposure despite the absence of PCR-confirmed active infection. This discrepancy, previously reported in LASV-endemic areas [10,13], may result from resolved infections with viral clearance or persistent low-level viremia below molecular detection thresholds [39,41,60]. While demonstrating exposure, seropositivity alone does not establish reservoir competence—the ability to maintain and transmit the virus. The consistent PCR-negative results across all samples indicate no active viral replication or shedding was occurring at the time of sampling, suggesting that these species may function as spillover hosts rather than primary reservoirs in this ecological context.

Notably, all seropositive animals were male, and *Rattus rattus* represented 75% of positive cases. While the small sample size limits definitive conclusions, this observation aligns with known ecological factors: male rodents typically exhibit larger home ranges and increased exploratory behavior, potentially enhancing contact rates with infected environments or conspecifics [61,62]. The concentration of seropositivity in synanthropic species, particularly *R. rattus*, supports emerging evidence that peri-urban LASV transmission cycles may involve non-Mastomys rodents [40,42,63]. The absence of significant spatial variation in seroprevalence across communities (ANOVA *p* > 0.05) suggests widespread low-level exposure throughout the study area.

The marginally significant difference in antibody detection between serum and plasma samples (χ^2^ = 3.29, *p* = 0.069) could be due to the anticoagulant in EDTA tubes potentially interfering with antibody–antigen binding in the ELISA, a phenomenon noted in other serological assays and thus underscores the importance of standardized sample processing protocols for serological sensitivity [64,65]. Although the chi-square test indicated only a marginal difference in infection rates between plasma and serum samples (*p* = 0.0696), this finding underscores the potential influence of sample type and handling conditions on serological sensitivity. Previous studies have shown that factors such as centrifugation procedures, storage temperature, and sample preservation duration can affect antibody stability and detection outcomes [63,64,66]. While these variables were not directly assessed in the present study, they remain important considerations for interpreting serological results and for designing future investigations. This technical observation, combined with the low prevalence rate, reflects the challenging nature of detecting intermittent LASV circulation in rodent populations, consistent with findings from other endemic regions [2,19,67,68,69,70,71,72,73,74,75,76].

The complete absence of *Mastomys natalensis*, coupled with the serological evidence in commensal species, suggests complex ecological interactions in peri-urban environments. While some studies suggest competitive exclusion might limit *Mastomys* establishment in human dwellings [2,17], the public health implications of alternative hosts require careful consideration. Rather than indicating “ecological protection,” the presence of LASV-exposed R. rattus and M. musculus may represent alternative transmission pathways deserving of further investigation.

Our findings reinforce that LASV epidemiology extends beyond simple host presence to include complex interactions between rodent ecology, human behavior, and environmental factors. While *M. natalensis* remains the established primary reservoir, multi-species surveillance approaches are essential to understand local transmission dynamics, particularly in rapidly evolving peri-urban landscapes where human-rodent interfaces are increasing.

### 4.4. Implications and Study Constraints

This study provides serological evidence of Lassa virus (LASV) exposure in *Rattus rattus* and Mus musculus within Otukpo LGA. However, key important limitations constrain the interpretation of these findings and highlight directions for future research.

Key constraints include the relatively small sample size of successfully processed animals (*n* = 92) and a high post-capture mortality rate (41.4%), which may have introduced survival bias and limited the statistical power of our analyses. The complete absence of *Mastomys natalensis*—the established primary LASV reservoir—from our captures prevents direct ecological comparison between reservoir and non-reservoir species in this setting. Furthermore, the lack of PCR-positive results, while consistent with patterns of transient or low-level viremia, precludes definitive conclusions about viral shedding or the reservoir competence of the seropositive animals.

These limitations significantly affected the scale and potential insights of this investigation. Future studies should prioritize optimized trapping, handling, and transport protocols to improve animal survival rates. Additionally, expanding surveillance to include diverse habitats—particularly rural and sylvatic zones where *Mastomys natalensis* is likely more prevalent—would enable a more comprehensive understanding of LASV host ecology and transmission dynamics across the epidemiological landscape.

Addressing these constraints and the complex ecology of LASV requires a One Health framework. This approach recognizes that the health of humans, domestic and wild animals, and the wider environment are inextricably linked. Moving beyond siloed efforts, a coordinated One Health strategy would be essential to unravel the transmission dynamics at the human-animal-environment interface in North-Central Nigeria.

## 5. Conclusions and Perspectives

This study confirms that LASV exposure occurs in some non-*Mastomys* rodents (*Rattus rattus*, *Mus musculus*) within the sampled peri-urban environments of Otukpo LGA. It thus provides critical baseline data on rodent and other small mammal diversity, survival, and Lassa virus (LASV) exposure in Otukpo LGA, North-Central Nigeria. The findings highlight the emerging role of *Rattus rattus* and *Mus musculus*, alongside *Mastomys natalensis*, as potential LASV reservoirs, emphasizing the need to broaden surveillance to include synanthropic and understudied species. The complete lack of captured *Mastomys natalensis* within this LF-endemic region is a surprising and significant finding, underscoring the need for improved sampling strategies targeting this primary reservoir for meaningful ecological context. Future investigations require larger-scale, longitudinal designs, incorporating diverse habitats and rigorous methods to ensure adequate survival and sample sizes of both target non-*Mastomys* species and *Mastomys natalensis* to robustly evaluate the potential contribution of alternative hosts to LASV ecology.

Challenges with live trapping and post-capture survival point to the urgency of refining capture protocols and improving animal care to ensure both ethical standards and data integrity. Enhancing sampling strategies, such as microhabitat targeting, increased trap density, and optimized baiting alongside longitudinal monitoring across ecological gradients, will provide deeper insights into host turnover, transmission dynamics, and LASV prevalence. To support effective LASV control, it is essential to expand surveillance of potential reservoirs and develop species-specific handling guidelines while integrating ecological, virological, and spatial data into predictive models. Strengthening local research capacity, fostering interdisciplinary collaboration, and promoting community-level awareness alongside rodent- and small-mammal-proofing initiatives will further contribute to reducing the risks posed by this virus across Nigeria and the wider West African UN subregion. At the same time, it is important to emphasize that the observed seropositivity in non-*Mastomys* rodents reflects prior exposure to LASV, rather than confirming these species as competent reservoirs. While such findings broaden our understanding of possible host diversity, they do not provide direct evidence of viral shedding or transmission dynamics. Follow-up studies focusing on active infection, viral load, shedding patterns, and transmission potential are therefore crucial to assess the epidemiological significance of these species and to inform more comprehensive and proactive strategies for Lassa fever prevention in West Africa.

## Figures and Tables

**Figure 1 viruses-17-01368-f001:**
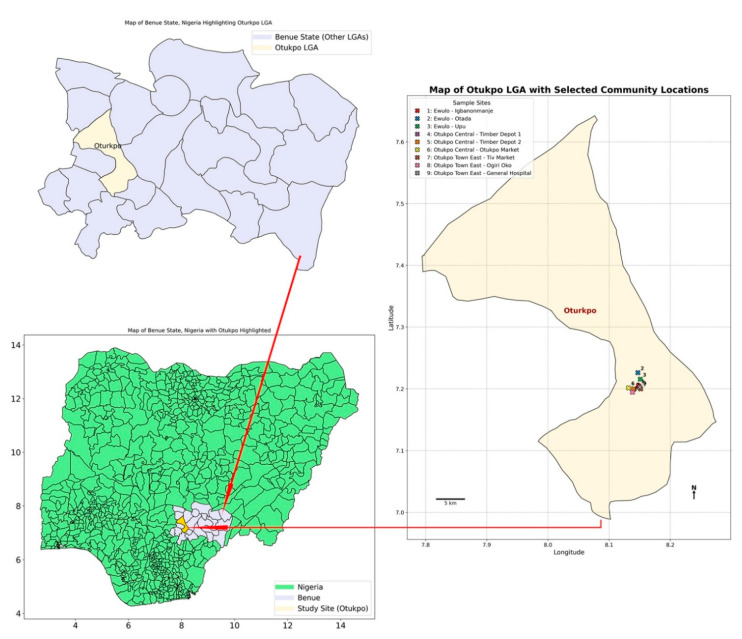
Map of Nigeria showing Benue State, Otukpo LGA and the wards/communities where small mammals were trapped.

**Figure 2 viruses-17-01368-f002:**
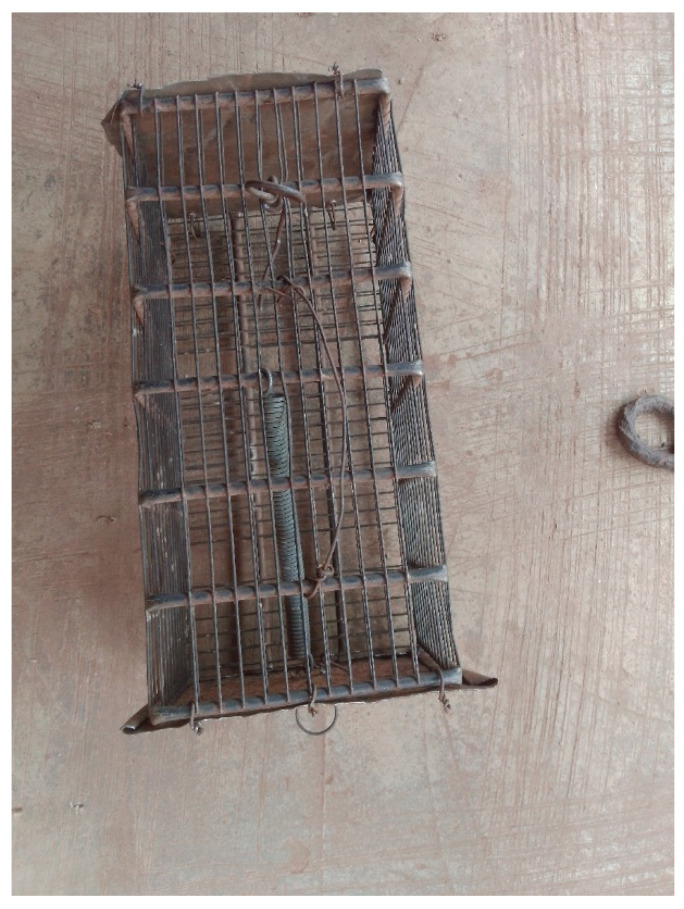
A fabricated live-capture trap used in the study.

**Figure 3 viruses-17-01368-f003:**
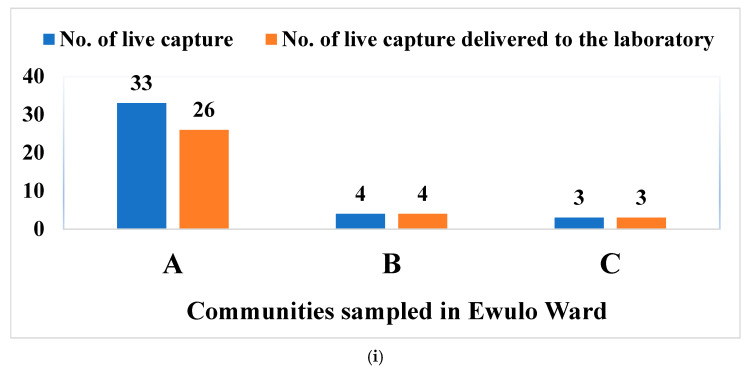
(**i**–**iii**) Live capture rate vs. survival rate of rodents and rats. **Note: LGA wards/communities: Ewulo Ward: A** = Igbanonmanje; B = Otada; C = Upu; Otukpo **Town Central Ward: D** = Timber Depot 1; **E** = Timber Depot 2; **F** = Otukpo main market axis; **Otukpo Town East Ward: G** = Tiv market axis; **H** = Ogiri Oko street; **I** = General Hospital axis.

**Figure 4 viruses-17-01368-f004:**
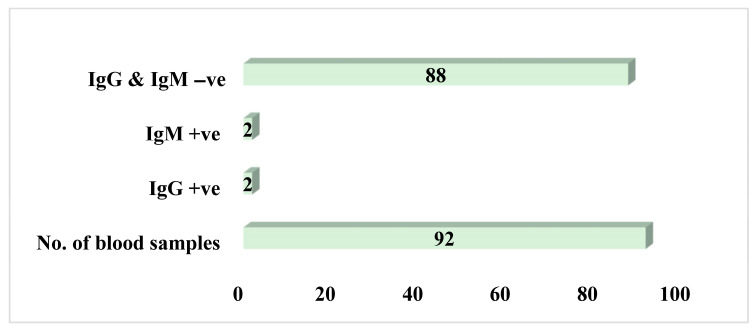
Summary of blood samples positive for both IgG and IgM.

**Figure 5 viruses-17-01368-f005:**
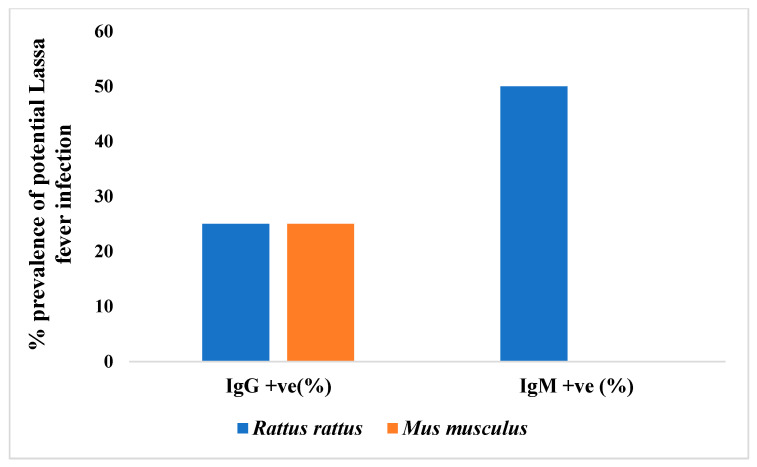
Percentage potential prevalence of Lassa fever infection among captured small mammals.

**Figure 6 viruses-17-01368-f006:**
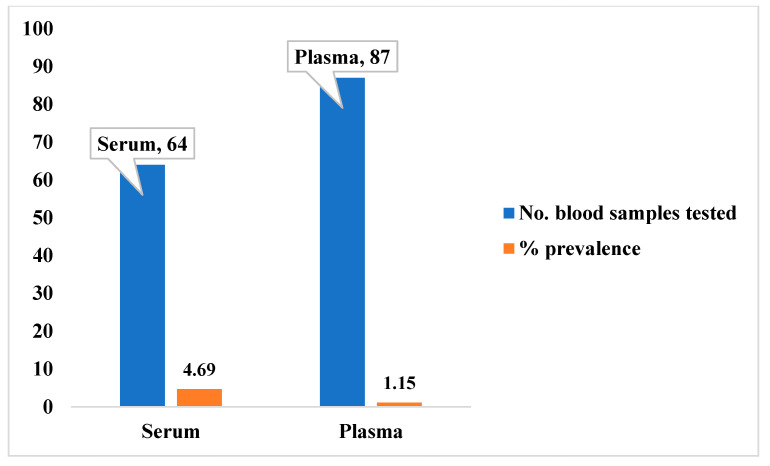
Number of positive cases of LASV antibodies in the blood samples of the small mammals among the communities sampled.

**Table 1 viruses-17-01368-t001:** Number of individuals of rodents and rats and capture success rate per sampling community.

Capture Rate (%) for:									
	Communities in Ewulo Ward			Communities in Otukpo Central Ward			Communities in Otukpo Town East Ward		
Rodents and Rats’ Taxa	A	B	C	D	E	F	G	H	I
*Mus musculus*	(9) 47.37%	(3) 12.50%	(1) 5.0%	(9) 45.0%	(7) 14.0%	(0) 0.00%	(0) 0.00%	(0) 0.00%	(0) 0.00%
*Rattus rattus*	(12) 63.16%	(1) 4.17%	(2) 10.0%	(15) 27.0%	(4) 8%	(15) 37.5%	(5) 66.67%	(1)5.6%	(2) 18.5%
*Rattus norvegicus*	(0) 0.00%	(0) 0.00%	(0) 0.00%	(0) 0.00%	(0) 0.00%	(1) 18.8%	(0) 0.00%	(0) 0.00%	(0) 0.00%
*Tatera* spp.	(4) 10.50%	(0) 0.00%	(0) 0.00%	(0) 0.00%	(0) 0.00%	(0) 0.00%	(0) 0.00%	(0) 0.00%	(0) 0.00%
*Pallasiomys* spp.	(1) 5.26%	(0) 0.00%	(0) 0.00%	(0) 0.00%	(0) 0.00%	(0) 0.00%	(0) 0.00%	(0) 0.00%	(0) 0.00%

**Note: LGA wards/communities: Ewulo Ward: A** = Igbanonmanje; **B** = Otada; **C** = Upu; **Otukpo Town Central Ward: D** = Timber Depot 1; **E** = Timber Depot 2; **F** = Otukpo main market axis; **Otukpo Town East Ward: G** = Tiv market axis; **H** = Ogiri Oko street; **I** = General Hospital axis.

## Data Availability

The data presented in this study are available on request from the corresponding author due to the funders requirement of holding the dataset for a period of two years before release to the general public until completion of the entire study.

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
