# Peer review of "Serological Evidence of Lassa Virus Exposure in Non-Mastomys Small Mammals Within a Hyperendemic Region of North-Central Nigeria: A Pilot Study"

_viruses, 2025, doi:10.3390/v17101368_

Round 1
Reviewer 1 Report
Comments and Suggestions for Authors
Comments to the Author
I congratulate the authors on this excellent work. In this manuscript, preliminary evidence for LASV exposure in non-Mastomys rodents is presented, underscoring the necessity of expanded surveillance.
I have some comments:
Major revisions
Q1. Supplement ecological data to explain the absence of Mastomys natalensis (e.g., habitat surveys) and discuss implications for study conclusions. For example, while the study sites are described as "high-risk areas," the absence of Mastomys natalensis is not adequately explained (e.g., habitat preferences, trapping method biases), undermining the ecological relevance of the conclusions.
Q2. Of the 157 total captures, only 58.6% (92 individuals) survived to laboratory testing, with significant survival differences across communities (ANOVA, P<0.05). However, the potential impact of survival bias on virus detection (e.g., stress-induced changes in viral load) is not analyzed. Additionally, the skewed sex ratio (65 males vs. 27 females) may introduce sampling bias, yet the study merely notes "no significant difference" without discussing implications for population structure analysis.
Q3. RT-PCR targets only the nucleoprotein (NP) gene, with no information on primer coverage of all LASV lineages, increasing the risk of false negatives. And The ELISA for LASV antibodies does not specify the antigen subtype used, despite known genetic diversity in LASV, which could affect detection specificity. Describe this further.
Q4. Temper conclusions to clarify that seropositivity indicates exposure, not reservoir competence, and propose follow-up studies on viral shedding and transmission dynamics.
Minor revisions
Q1. In 2.3 section, while trap dimensions and materials are specified, the type of traps (live vs. lethal) and their impact on sample survival are not described. Trap density (e.g., per hectare) is omitted, hindering assessment of trapping efficiency.
Q2. Results line 365: It is stated that the capture rate of Rattus rattus in Timber Depot 2 is 80%. However, combined with the information in Table 1 (only 4 individuals of R. rattus were captured in this community), it was unclear whether the capture rate is 8% or 80%. The original data needs to be checked for verification.
Q3. Figures 6 and 7 redundantly presented seroprevalence differences between serum and plasma. These should be merged to emphasize the core finding that serum (4.69%) yields higher positivity rate than plasma (1.15%).
Q4. Discussion lines 502-504: A chi-square test indicated a "marginally significant" difference in positivity rates between serum and plasma (P=0.0696), but the study does not investigate how sample processing (e.g., centrifugation, storage temperature) might influence results, relying instead on vague assertions about "sample type and storage conditions."
Q5. References require standardization (e.g., Reference 48).
Q6. The figures presented the necessary data, but improving their visual presentation (e.g., through consistent formatting, clearer labeling, and refined design elements) would enhance their clarity and effectiveness in communicating your findings (e.g., plate 1).

Reviewer 2 Report
Comments and Suggestions for Authors
This pilot study investigates the presence of Lassa virus (LASV) exposure in non-Mastomys small mammals in Otukpo LGA, Benue State, Nigeria, and to assess their prevalence and potential role in LASV transmission.
The authors have clearly stated some of the implications and constrains of the study (section 4.4.). Logistical challenges such low rate of survival and other findings like the absence of M. natalensis, and serological evidence of exposure, but no active infection detected via PCR. These limitations preclude a generalization of the results, but their findings remain interesting for a pilot study.
For future work, I would suggest that apart from the obvious optimizing of the trapping and handling protocols for the animals, to expand surveillance to include diverse habitats and species. Last, it would be very useful to be able to integrate ecological, virological, and spatial data for better predictive models.
Other comments:
-Authors should make proper reference in text to “Plate 1” figure and follow same figure naming convections along all manuscript.
-Why are traps described in terms of their diameter ?
-Section 3.1. improve figure description (e.g. fig 2.ii C,D,E not referenced in the text)
Comments on the Quality of English Language
-In general, the methods section could use some improvements in writing.
